# Identifying New Contributors to Brain Metastasis in Lung Adenocarcinoma: A Transcriptomic Meta-Analysis

**DOI:** 10.3390/cancers15184526

**Published:** 2023-09-12

**Authors:** Vanessa G. P. Souza, Aisling Forder, Nikita Telkar, Greg L. Stewart, Robson F. Carvalho, Luis A. J. Mur, Wan L. Lam, Patricia P. Reis

**Affiliations:** 1Molecular Oncology Laboratory, Experimental Research Unit (UNIPEX), Faculty of Medicine, São Paulo State University (UNESP), Botucatu 18618-687, SP, Brazil; 2British Columbia Cancer Research Institute, Vancouver, BC V5Z 1L3, Canada; aforder@bccrc.ca (A.F.); ntelkar@bccrc.ca (N.T.); gstewart@bccrc.ca (G.L.S.); wanlam@bccrc.ca (W.L.L.); 3British Columbia Children’s Hospital Research Institute, Vancouver, BC V5Z 4H4, Canada; 4Department of Structural and Functional Biology, Institute of Biosciences, São Paulo State University (UNESP), Botucatu 18618-689, SP, Brazil; robson.carvalho@unesp.br; 5Department of Life Science, Aberystwyth University, Aberystwyth, Wales SY23 3FL, UK; lum@aber.ac.uk; 6Department of Surgery and Orthopedics, Faculty of Medicine, São Paulo State University (UNESP), Botucatu 18618-687, SP, Brazil

**Keywords:** lung cancer, bioinformatics, brain metastasis, immune cell, tumor microenvironment (TME)

## Abstract

**Simple Summary:**

Lung cancer patients have a high mortality risk due to brain metastases (BM). Understanding the molecular changes that contribute to BM is essential to identify potential therapeutic targets. Previous research has focused on primary tumor alterations, with less attention given to BM. This study examined a unique transcriptomic dataset assembled from previously reported RNA-seq, microarray, and single-cell analyses of BM samples from lung adenocarcinoma (LUAD) patients in pursuit of gaining a better understanding of the molecular landscape of BM. We found that dendritic cells and neutrophils were present in LUAD-BM, which could contribute to an immunosuppressive tumor microenvironment. The expression levels of 102 genes were altered, with *CD69* and *GZMA* identified as ‘hub’ genes, which could play a role in LUAD-BM. BM-specific gene expression was also observed, further supporting the presence of an immunosuppressive tumor microenvironment.

**Abstract:**

Lung tumors frequently metastasize to the brain. Brain metastasis (BM) is common in advanced cases, and a major cause of patient morbidity and mortality. The precise molecular mechanisms governing BM are still unclear, in part attributed to the rarity of BM specimens. In this work, we compile a unique transcriptomic dataset encompassing RNA-seq, microarray, and single-cell analyses from BM samples obtained from patients with lung adenocarcinoma (LUAD). By integrating this comprehensive dataset, we aimed to enhance understanding of the molecular landscape of BM, thereby facilitating the identification of novel and efficient treatment strategies. We identified 102 genes with significantly deregulated expression levels in BM tissues, and discovered transcriptional alterations affecting the key driver ‘hub’ genes *CD69* (a type II C-lectin receptor) and *GZMA* (Granzyme A), indicating an important role of the immune system in the development of BM from primary LUAD. Our study demonstrated a BM-specific gene expression pattern and revealed the presence of dendritic cells and neutrophils in BM, suggesting an immunosuppressive tumor microenvironment. These findings highlight key drivers of LUAD-BM that may yield therapeutic targets to improve patient outcomes.

## 1. Introduction

Brain metastases (BM) are a common and serious complication in patients with lung cancer, with tumors of the lung being the most prevalent cause of brain metastases [1,2,3,4,5]. The incidence of BM in lung cancer varies according to tumor histology where non-small-cell lung cancer (NSCLC), which accounts for approximately 85% of all lung cancer cases, has an incidence of BM ranging from 10 to 50% [6,7]. Lung adenocarcinoma (LUAD) is the predominant histological subtype of NSCLC, and represents approximately 40% of all lung cancer cases. LUAD is frequently diagnosed at an advanced stage and characterized by the presence of metastases, with the brain being one of the main metastatic sites [8].

BM is associated with a wide range of symptoms, including headaches, seizures, and changes in vision, speech, and/or behavior [9]. These symptoms significantly impair patient quality of life, and the presence of LUAD-linked BM is associated with a dismal prognosis [10] and a median survival of approximately 15 months [10]. Treatment options for BM are limited but may include surgery, radiation therapy, and chemotherapy: ultimately, the choice of treatment depends on multiple factors including the patient’s overall health and the extent of the metastasis [9,11].

Recent advances in targeted therapies and immunotherapies have shown promise in treating lung cancer, including BM. However, it is essential to conduct more research to improve outcomes since treating BM remains a persistently serious and difficult challenge [12,13,14,15]. Previous investigations have mainly focused on studying primary tumors with and without metastasis, in order to shed light on the underlying mechanisms of BM in lung cancer patients, paving the way for developing treatment strategies [16,17,18,19,20]. The molecular landscape of lung cancer-related BM has been recently reviewed comprehensively [11]. There is an urgent need for more research, since most studies are limited by small sample sizes and include patients with different disease subtypes lacking a complete description of patient clinical data. Therefore, further investigations are warranted, as the previous data have not been translated into clinical practice to provide discernible benefits to patients.

At present, targeted drugs available for the treatment of BM in lung cancer only benefit a subset of patients, are very costly, and associated with toxicity and development of resistance [12,13]. Therefore, it is of utmost importance to perform large-scale molecular studies of BM tissues to develop biomarker-based therapies. Genomic analyses of BM and corresponding primary tumors and other extracranial metastases have revealed that BM may harbor potentially actionable driver mutations [21]. The identification of specific molecular targets for BM will likely contribute to improved outcomes of patients who develop BM.

Here, we integrated data from different sequencing technologies (bulk RNA-seq, microarray, and single-cell RNA-seq) to provide a more comprehensive and detailed understanding of the molecular mechanisms and microenvironment components involved in BM from LUAD. In particular, we aimed to unravel transcriptomic changes that were specific to metastatic tumor cells within the brain, which might differ from the primary tumor. Furthermore, we employed CIBERSORTx [22], a widely utilized analytical tool in tumor immunity research, to extract information on cell subsets from bulk gene expression data, with the objective of identifying potential biomarkers for BM and evaluating the presence of immune cells in LUAD patients’ BM tissues. Finally, we sought to determine immune-related genes that could be applicable for diagnosing and treating brain metastasis by calculating the correlation between immune cells and ‘hub’ genes. This integrated approach holds promise in identifying novel therapeutic targets and fostering the development of more precise treatment approaches for this complex disease. A schematic overview of the methodology employed in our study is shown in Figure 1.

## 2. Materials and Methods

### 2.1. RNA-Seq and Microarray Data Selection for Meta-Analysis

We searched for brain metastasis from lung adenocarcinoma-related RNA-seq and microarray datasets in different public repositories and databases using the following search terms: “brain metastasis”, “brain metastasis AND lung adenocarcinoma”, “brain metastasis AND transcriptome”, “brain metastasis AND transcriptomics”, “brain metastasis AND microarray”, ”brain metastasis AND RNA-seq”. We included eight databases or repositories: Human Cancer Metastasis Database (HCMDB) [23], ArrayExpress [24], Restructured GEO (ReGEO) [25], European Genome-phenome Archive (EGA) [26], NCI Genomic Data Commons (GDC) [27], Sequence Read Archive (SRA) [28], The Database of Genotypes and Phenotypes (dbGaP) [29,30] and Gene Expression Omnibus (GEO) [31]. The search was performed on 12 September 2021. We selected the datasets using the following criteria: (1) gene expression data from BM tissue of patients diagnosed with LUAD as the primary tumor, (2) gene expression data from the primary tumor tissue of LUAD patients diagnosed with BM, (3) all BM locations were considered, (4) all platforms were considered. Exclusion criteria were: (1) leptomeningeal metastases, (2) studies that used only cell lines or animal models, and (3) review studies. The transcriptomic data used in this study were divided according to the sequencing technology: RNA-seq (*n* =13 BM; 11 Primary tumor), and microarray (*n* = 63 BM; 77 Primary tumor).

### 2.2. Identification of Differentially Expressed Genes Using RNA-Seq Data

The bam or fastq files were obtained from different repositories. The datasets obtained from SRA were downloaded using the SRA Toolkit (v.2.8.0) (available online at: https://www.ncbi.nlm.nih.gov/sra/docs/sradownload/, accessed on 1 October 2021) and converted from sra format to fastq using the fastq-dump --split-3 identifier. The datasets obtained from EGA were access-controlled; therefore, for each dataset, access was required through the Data Access Committees (DAC), providing documentation of the data access agreement. Once access was authorized, the bam files were downloaded using the EGA download client tool [32]. The arguments used were pyega3 -cf </Path/To/CREDENTIALS_FILE> datasets. The bam files were converted to fastq using Samtools [33]. The quality of the data was initially assessed with FastQC (v.0.11.9) [34] and summarized with MultiQC (v.1.10) [35]. All the reads were filtered by quality in SeqyClean (v.1.10.09) [36] using Phred (QS) 30 and 30 for the mean and edge minimum score values and a minimum length of 65 bp. SeqyClean was also used to remove contaminant sequences from primers and vectors using the Univec database [37]. The reads were aligned with the Ensembl human genome assembly GRCh38 (release 99) using STAR 2-pass alignment method (v.2.7.8a) [38], and the expression count matrix was generated using HTSeq (v.0.11.1) [39]. Qualimap (v.2.2.1) [40] was used for quality control of the sequence alignment. Combat-Seq (v.3.36.0) [41,42] was used to remove batch effects between datasets. The EdgeR package was used to identify differentially expressed genes (DEGs) between BM and primary tumor. The *p*-value was adjusted by the Benjamini–Hochberg method to control the false discovery rate (FDR). Genes with the cutoff criteria of |logFC| > 2 and adj. *p* < 0.05 were considered DEGs. DEGs were visualized as a volcano plot using the package (v.3.3.5) [43]. The gplots (v.3.1.1) [44], and biomaRt (v.3.13) [45,46] packages were used to build the heatmap and convert the ensembl_gene_id to hgnc_symbol, respectively.

### 2.3. Identification of Differentially Expressed Genes Using Microarray Data

Microarray data were obtained from the Gene Expression Omnibus (GEO) and ArrayExpress public databases. The E-MTAB-8659 dataset obtained from ArrayExpress based on the Illumina HumanHT-12 V4.0 expression beadchip platform included 63 brain metastasis samples from patients diagnosed with adenocarcinoma as the primary tumor. Additionally, we selected a GEO dataset (accession: GSE60645) that included 77 tissue samples from the primary LUAD tumor profiled using the Illumina HumanHT-12 V4.0 expression beadchip platform (there is no information about the presence or absence of BM in these data). Only datasets generated from the Illumina HumanHT-12 V4.0 expression beadchip platform were processed in order to minimize cross-platform variation. The microarray datasets were processed and normalized using *limma* (v.3.50.0). After normalization, *limma* [47] was used to identify DEGs between BM and the primary tumor. FDR value < 0.05 and |logFC| > 1.5 were used as cutoff criteria for DEGs. The DEGs were visualized as a volcano plot using the ggplot2 package (v.3.3.5) [43]. The gplots package (v.3.1.1) was used for building the heatmap [44].

### 2.4. Identification of Differentially Expressed Genes (DEGs) Overlap between RNA-Seq Data and Microarray

To identify common DEGs between RNA-seq data and microarray data, only transcripts with HGNC-approved nomenclature were considered (available online at: www.genenames.org, accessed on 15 October 2021). The HGNC is responsible for approving unique symbols and names for human loci, including protein-coding genes, noncoding RNAs, and pseudogenes, to facilitate an unambiguous report. HGNC generally does not assign gene symbols to transcripts alternative or *splice* variants [48]. Genes common between both technologies were presented as a Venn diagram using the VennDiagram (v.1.7.1) [49]. Only the genes with consistent direction of expression change among the sequencing technologies were considered.

### 2.5. Functional and Pathway Enrichment Analyses

In order to obtain information about the biological function of the identified genes, we performed functional annotation and pathway enrichment analyses. To explore Gene Ontology (GO), we used the enrichGO() function from the R package clusterProfiler (v. 4.0.5) [50]. Additionally, we simplified the GO enrichment output by removing the redundancy of enriched GO terms using the simplify() function. The GO annotation file for the human species was obtained from the Gene Ontology (available online at: http://geneontology.org/, accessed on 1 October 2021) [51,52]. For the Kyoto Encyclopedia of Genes and Genomes (KEGG) enrichment analysis, we used the enrichKEGG() function, also from clusterProfiler. The DOSE package (v.3.14) from Bioconductor was used for disease ontology enrichment analysis based on the Disease Ontology (DO) database (available online at: https://disease-ontology.org/, accessed on 1 October 2021) [53], as well as enrichment analysis based on the Network of Cancer Genes (NCG) database (available online at: http://ncg.kcl.ac.uk/index.php, accessed on 1 October 2021) [54]. For these analyses, the enrichDO(), and enrichNCG() functions were used, respectively. In all analyses, the *p*-value was adjusted using the Benjamini–Hochberg method to control the false discovery rate (FDR). Categories with a cutoff of *p*. adj < 0.05 were considered significant. Ggplot2 and GOplot packages were used to visualize the results [43,55].

### 2.6. Protein–Protein Interaction Network Construction for Selected Modules and Hub Genes Identification

Gene symbols for the common DEGs were converted to UniProt IDs using the org.Hs.eg.db (v.3.17) package [56]. Then, they were analyzed by the online tool STRING [57] for the construction of a Protein–protein interaction (PPI) network. Active interaction sources, including text mining, experiments, databases, co-expression, species limited to “Homo sapiens” and an interaction score  >  0.4 were applied to construct the PPI networks. The results were visualized with CytoScape software (v.3.10.0) [58]. CytoHubba, a plug-in of CytoScape, was used to identify the PPI network’s central elements [59]. Genes with the top 20 maximal clique centrality (MCC) values were considered ‘hub’ genes. The adjusted *p*-values (Benjamini–Hochberg method) were deemed significant at *p* < 0.05.

### 2.7. Immunophenotype of Brain Metastasis from Lung Adenocarcinoma

To estimate the immunological composition of the samples, we used the analytical tool CIBERSORTx [60]. CIBERSORTx includes the LM22 file, a signature matrix composed of 547 genes that distinguish 22 mature human hematopoietic populations, including seven types of T cells, B cells, plasma cells, NK cells, and myeloid subsets. Before being submitted to CIBERSORTx, the raw count data from RNA-seq was transformed into transcripts per million kilobases (TPM) using the convertCounts() function of the R package DGEobj.utils (v.1.0.4) [61]. The microarray data were processed by the *limma* package and used to validate the findings obtained with RNA-seq data. All samples were analyzed for immune cell profiles using CIBERSORTx with the number of permutations set to 1000 in order to obtain high statistical accuracy, and quantile normalization was turned off for RNA-seq data. The output files were downloaded as tab-delimited text files and imported into the R software (v.4.1.1) [62], which was used to identify differences in immune composition between BM and T. The normality test was performed using the shapiro.test() function. Differences were considered significant when *p* < 0.05 by the Wilcoxon–Mann–Whitney test. Then, we explored the association between the populations of immune cells in the studied groups; for this, the Spearman correlation analysis was calculated using the rcorr() function of the Hmisc package (v.4.6-0) [63]. Additionally, we analyzed the correlation between the infiltration of the 22 cell types of the immune system and the expression of the ‘hub’ genes. The function chart.Correlation() from the PerformanceAnalytics package (v.2.0.4) [64] was used to obtain the expression scatter plots of the ‘hub’ genes along with Spearman correlation and estimated statistical significance. Values were considered significant when *p* < 0.05. Gene expression levels were determined with log2 TPM. The heat and chord plots were generated using the packages ggplot2 (v.3.3.5) [43], and circlize (v.0.4.13) [65].

### 2.8. Single-Cell RNA-Sequencing Data Processing and Analysis

Single-cell RNA sequencing (scRNA-seq) data for LUAD-BM samples were downloaded from Gene Expression Omnibus [31] (GSE131907, *n* = 10; GSE143423, *n* = 3; GSE202371, *n* = 10). Seurat (v.4.0.2) [66,67] was used for data quality control, integration, and analysis. Briefly, Seurat objects were created from individual expression matrices. Cells expressing <200 or >9000 genes (outliers) or with a percentage of mitochondrial genes higher than 10%, and genes expressed in less than 3 cells were all excluded (Appendix A). For the remaining cells, gene expression matrices were normalized using the NormalizeData function in the Seurat package. Seurat FindVariableFeatures were applied to select the top 2000 genes exhibiting the highest cell-to-cell variation. Gene expression matrices from different samples were then integrated. The batch effects were removed by canonical correlation analysis and mutual nearest neighbors-anchors using the functions SelectIntegrationFeatures, FindIntegrationAnchors, and IntegrateData. Subsequently, the integrated, batch-corrected expression matrix for all cells was scaled with the Seurat ScaleData function to apply a linear transformation. Principal component analysis (PCA) and uniform manifold approximation and projection (UMAP) were used for dimensionality reduction. We determined the dimensionality of the dataset using the JackStrawPlot function. The top 15 principal components (PCs) were selected for dimensionality reduction. Before clustering the cells, a shared nearest neighbor graph based on the Euclidean distance in PCA space was conducted using Seurat FindNeighbors. Clustering was then performed with Seurat FindClusters with a resolution of 1.2. Marker genes for each cluster were determined with the Wilcoxon rank-sum test using Seurat FindAllMarkers. For each cluster, only genes that were expressed in more than 25% of cells with at least 0.25-fold difference were considered. The annotations of cell identity on each cluster were defined by the expression of known marker genes.

## 3. Results

### 3.1. Datasets Selected for Meta-Analysis

We performed an extensive search in different public databases that contain transcriptomic data using microarray platforms and/or high-performance sequencing (RNA-seq). The search resulted in six studies: four reporting gene expression data from BM tissue of patients diagnosed with LUAD as the primary tumor; one study reporting gene expression data from primary tumor tissue of LUAD patients diagnosed with BM; and one study with both of the above information. The description of the publicly available studies is shown in Table 1. Dataset BioProject: EGAD00001007973 was excluded from our analyses as we were not able to access the raw data. Therefore, transcriptome expression data from five studies was included in our analysis, four with RNA-seq data and one with microarray data. Additionally, we selected a dataset (accession: GSE60645) based on the Illumina HumanHT-12 V4.0 expression beadchip platform that included 77 tissue samples from the primary LUAD tumors. The clinical information for the studies included in the meta-analysis is provided in Appendix A.

### 3.2. Integration of RNA-Seq and Microarray Datasets Identified 102 Differentially Expressed Genes in Brain Metastasis from Lung Adenocarcinoma

After quality control, mapping, and data normalization (Appendix A), we proceeded with differential expression analysis. A total of 164 samples (88 primary tumors and 76 BM) were included for differential expression analysis (Figure 1). Analysis was performed using the R programming language using two packages: *edgeR* and *limma*. Due to the large size of the tested genes, raw *p*-values were adjusted according to Benjamini and Hochberg’s method for false discovery rate correction. The selection criteria were strengthened with a threshold of |logFC| > 2 and adj. *p* < 0.05 for RNA-seq and FDR value < 0.05 |logFC| > 1.5 for microarray data. These thresholds were chosen to detect significant gene expression changes against the inherent technical and biological variation within each platform. Volcano plots were generated to illustrate the distribution of each gene according to the logFC and adjusted *p*-value (Figure 2). In RNA-seq data, these parameters generated a list of 6426 differentially expressed genes (DEGs) in BM in comparison to the primary tumor, with 1850 upregulated and 4576 downregulated genes (Figure 2A) (Appendix A). Within the microarray data, 268 genes were significantly differentially expressed in BM in comparison to the primary tumor, with 18 upregulated and 250 downregulated genes (Figure 2B) (Appendix A). Interestingly, among the sequencing technologies, 106 DEGs (1.58%) (Appendix A) were overlapping between RNA-seq and microarray, while 102 DEGs (1.52%) showed the same expression direction between the two sequencing platforms (Figure 3A) (Appendix A). *JSRP1*, *CAMK1G*, *COX7A1*, and *NCALD* did not show agreement in the direction of gene expression changes identified between sequencing technologies and were thus removed from subsequent analyses (Figure 3B).

### 3.3. Pathway Enrichment Analysis Showed Pathways in BM Are Associated with the Immune System

To obtain information about functional mechanisms regulated by the 102 DEGs, analyses of functional annotation and pathway enrichment were performed. In the analysis of the Kyoto Encyclopedia of Genes and Genomes (KEGG) pathway, enrichment of the following molecular pathways was identified: cell adhesion, chemokine signaling, cytokine–cytokine receptor interaction, and Th1, Th2, and Th17 cell differentiation pathways (Figure 4A) (Appendix A). The dysregulated expression of these genes may play a crucial role in modulating immune responses and cell-to-cell communication processes. Furthermore, functional enrichment analysis was conducted to predict the biological functions associated with the DEGs. This analysis identified biological processes related to the immune response, chemokine response, and extracellular matrix organization. These findings suggest that the DEGs may be involved in immune-related processes. Moreover, the enrichment of cellular component terms mainly associated with the cell membrane indicates that the DEGs may have important roles in membrane-associated functions and signaling processes (Figure 4B) (Appendix A).

### 3.4. Brain Metastasis from Lung Adenocarcinoma Exhibits Distinctive Characteristics That Distinguish It from All Other Types of Cancer

In order to gain insights into the functional mechanisms regulated by the DEGs, we conducted analyses of functional annotation. One aspect we explored was the relationship of the DEGs in the context of diseases. To achieve this, we utilized the DOSE package [73] for disease ontology enrichment analysis. Disease ontology provides a framework for annotating human genes within the context of specific diseases, facilitating the translation of molecular findings into clinical relevance. Using gene set enrichment analysis, we identified significant associations between the DEGs and interstitial lung disease. Specifically, out of the 102 DEGs, nine genes (*CTSK*, *COL1A2*, *CCL5*, *AEBP1*, *CXCL9*, *CXCL10*, *CCL18*, *PDGFRA*, *CCL19*) exhibited significant associations with this condition (Figure 5A) (Appendix A). We also identified significant associations between the DEGs and bacterial infection disease (*TF*, *HLA-DPB1*, *CD247*, *CCL5*, *CD27*, *HLA-DQA1*, *GZMA*, *CXCL9*, *CXCL10*, *VCAM1*, *SH2D1A*, and *CCR7*).

Additionally, we performed an enrichment analysis based on the Network of Cancer Genes database to further explore the relationship between the DEGs and specific types of cancer. Surprisingly, our results did not reveal any significant associations between the DEGs and particular cancers, indicating that metastatic brain tumors possess unique characteristics that distinguish them from other types of cancer (Figure 5B) (Appendix A). These findings highlight the distinct molecular features and underlying mechanisms of metastatic brain tumors.

### 3.5. Protein–Protein Interaction Network Constructed from DEGs Reveals the Biological Network of Brain Metastasis from Lung Adenocarcinoma Is Associated with the Immune System

In order to understand the functional relationship of DEGs and the biological phenomena involved, we investigated the functional interactions of the proteins encoded by these genes through the construction of a connectivity network using the database STRING. This tool allows for achieving a comprehensive and objective global network, including direct (physical) and indirect (functional) protein interactions. The network of genes related to BM (Figure 6A) has 101 nodes. The network nodes represent proteins (each node represents all proteins produced by a single protein-coding gene locus) and 279 edges (edges represent protein–protein associations). Regarding the centralities of the network, the network presents the average of the local clustering coefficient = 0.509 and average degree = 5.52. The clustering coefficient is a measure of how connected the network nodes are. Highly connected networks have values close to 1. The average degree of a node is the number of how many interactions a protein has on average in the network and indicates the regulatory relevance of this protein. The PPI enrichment value was *p* < 1.0 × 10^−16^. This indicated that the proteins are biologically significantly connected. The interaction with the highest combined score was between the CD3D and CD247 proteins (combined score = 0.999) (Appendix A). The combined score is a confidence indicator, that is, the probability that the STRING considers an interaction to be true, according to the evidence available. All scores are ranked from 0 to 1, with 1 being the highest possible confidence.

After building the PPI network, we built a co-expression network using the maximal clique centrality (MCC) algorithm from the CytoHubba plug-in from Cytoscape. The tool allows for inferring the importance of nodes and helps to identify the central elements of a biological network. The MCC method classifies nodes (proteins) into high- and low-grade categories. The protein grade is a measure that indicates the degree of correlation between the protein and the essentiality of its corresponding gene, i.e., proteins with higher grades are more likely to be essential proteins in the biological network. Based on this analysis, we identified 20 key network elements (Figure 6B), referred to as ‘hub’ genes. The ‘hub’ genes and the score grades are shown in Appendix A. Among these, the CD69 gene showed the highest degree of connectivity (score = 396,192). Notably, all of these genes were downregulated (Figure 6C).

Related to the ‘hub’ gene enrichment analysis, key enriched terms of the biological process include T-cell activation, chemokine response, and upregulation of cell–cell adhesion (Appendix A). For enrichment analysis of cell components, the results are related to cell membrane components. The corresponding molecular function terms are represented in related terms, mainly chemokine activity (Appendix A). In the enrichment analysis of the Kyoto Encyclopedia of Genes and Genomes (KEGG) pathway, the main pathways identified were T-cell activation, Th1 and Th2 cell differentiation, and the TNF signaling pathway (Appendix A).

### 3.6. The Fraction of Neutrophils Is Greater in Brain Metastasis Compared to the Primary Tumor

Considering that the tumor immune microenvironment is known to play an important role in metastatic progression, and that our DEG analysis revealed enrichment of immune-related pathways and processes in LUAD-BM, we investigated the composition of immune cell infiltrates in the BM samples using the CIBERSORTx algorithm. Figure 7A summarizes the results obtained from the analysis of 13 BM samples and 11 primary tumors (RNA-seq data). In both groups, resting memory CD4 T cells comprised the largest cellular fraction of the total immune cells, with 17.4% of the total immune cells in BM and 15.24% in primary tumors (Figure 7A). After estimating the composition of immune cell infiltrates, we identified significant variations between the studied groups. We found that the fraction of resting dendritic cells (also referred to as immature dendritic cells) was significantly lower in BM compared to the primary tumor (T); while the neutrophil fraction was higher in the BM compared to the primary tumor (T) (Figure 7B,C).

We also explored the correlation between 22 immune cell subtypes in BM and the primary tumor by Spearman’s correlation (Appendix A, respectively). We identified several highly positive relationships between infiltrating immune cells in BM samples, while the mutual relationship between immune cells was reduced in primary tumor samples (Figure 8A,B). In BM, the highest positive correlation was between follicular T helper cells and plasma cells (Rho = 0.77, *p*-value = 0.001); while in primary tumors, the highest positive correlation was between plasma cells and monocytes (Rho = 0.80, *p*-value = 0.002). We further explored the correlation between the infiltration of the 22 cell types of the immune system and the expression of the 20 previously identified ‘hub’ genes in the BM samples (Figure 8C). The *CD27*, *CXCL13*, and *CD79B* were the genes that showed the highest number of correlations between their expression and the infiltration of immune cells, a total of seven significant correlations were identified for each of these genes. All genes were significantly correlated with the expression of CD8 T cells, naive CD4 T cells, monocytes, M1 macrophages, and resting mast cells; all correlations were positive except for resting mast cells (Rho = −0.77; −0.65 and −0.63, *p*-value < 0.05) and naive CD4 T cells (Rho = −0.56; −0.76 and −0.68, *p* < 0.05). Related to immune cells, CD8 T cells, naive CD4 T cells, regulatory T cells, and monocytes were the cell subtypes that showed the highest number of correlations significant with the expression of the ‘hub’ genes. Monocyte infiltration was correlated with the expression of 18/20 ‘hub’ genes (*CD69*, *CCR7*, *CD27*, *CD2*, *CCL5*, *CD247*, *GZMA*, *CD3D*, *GZMK*, *IL2RB*, *CXCL9*, *CCL19*, *CXCL13*, *CXCL10*, *CD48*, *VCAM1*, *CD79B* and *SLAMF6*), followed by CD8 T-cell infiltration correlated with expression of 16/20 ‘hub’ genes (*CD69*, *CD27*, *CD2*, *CCL5*, *CD247*, *GZMA*, *CD3D*, *GZMK*, *IL2RB*, *CXCL9*, *CCL19*, *CXCL13*, *CXCL10*, *CD48*, *CD79B* and *SLAMF6*). Naive CD4 T cells and regulatory T cells were both correlated with the expression of 15 ‘hub’ genes (*p* < 0.05). Therefore, ‘hub’ genes were correlated with immune-infiltrated cells in BM from LUAD.

### 3.7. scRNA-Seq Data Reveals an Immunosuppressed Tumor Microenvironment in BM from Lung Adenocarcinoma

To explore the function of neutrophils and dendritic cells in LUAD-BM we obtained a total of 57,774 cells from three different datasets (GSE131907, *n* = 24,508 cells; GSE143423, *n* = 12,196 cells; GSE202371, *n* = 21,070 cells) (Figure 9A). We applied the uniform manifold approximation and projection (UMAP) method and successfully classified the cells into 43 separate clusters (Figure 9B). Sub-clustering of 13,427 dendritic cells was identified, as shown in Figure 9C (Appendix A). For a more comprehensive analysis, we reclassified DCs into six subsets using markers previously described [74,75]. These subsets included CD1c+ DCs (Langerhans cells, LCs), CD141+ DCs, CD207 + CD1a+ LCs, pDCs (plasmacytoid DCs), CD163 + CD14+ DCs, and activated DCs (Figure 4D). Interestingly, a subset of activated DCs was not identified (Appendix A), only CD163 + CD14+ DCs were found (Figure 9D). The cluster of neutrophils containing 3762 cells was identified as polymorphonuclear myeloid-derived suppressor cells (PMN-MDSCs) (Figure 9). We next carried out a pathway analysis which showed that PMN-MDSCs was enriched in the IL-17 signaling pathway and NF-kappa B signaling pathway (Figure 9E) (Appendix A). Then, we found that CD163 + CD14+ DCs showed increased expression of HLA genes and antigen processing and presentation pathway (Figure 9F). They also showed high expression of *CCL3*, *CCL2,* and *CXCL3* (Appendix A*)*, which might be involved in the recruitment of activated T cells to inflammation sites [76].

## 4. Discussion

In the present study, we conducted a comprehensive search of public databases to obtain transcriptomic data of BM from patients diagnosed with adenocarcinoma as the primary tumor (LUAD-BM). Differential expression analysis is the most commonly used method for identifying genes expressed aberrantly in the context of interest. These differentially expressed genes are then explored through functional annotation analyses and enrichment of specific deregulated pathways. Despite the clear convenience of the approach, it is limited by a high level of noise in gene expression data, difficulty in the reproducibility of results, and individual differences due to factors such as age, sex, genotype, and disease stage. Additionally, different treatment stages and variations in cohort and experimental methods can also result in disparities between studies. Therefore, combining multiple studies represented a powerful strategy to address these issues and extract relevant information from various datasets. The variability in transcriptome analysis technologies (RNA-seq and arrays) expands transcriptional profiles, increasing the number of possibilities for identifying key molecular pathways associated with BM in lung adenocarcinoma.

As can be observed in Table 1, most studies have a limited number of samples, and therefore, a comprehensive and integrative analysis of these data can reveal new molecular components that are not identified when these studies are analyzed individually. We recently comprehensively reviewed the transcriptomic changes that are associated with the development of LUAD-BM, including alterations in gene expression in both coding and non-coding RNAs [11]. Here, by combining multiple datasets, we have systematically identified consistently altered expression levels of 102 genes across a larger number of BM from LUAD. Remarkably, the majority of differentially expressed transcripts showed decreased expression in BM. Although our meta-analysis identified mainly protein-coding genes, noncoding RNAs, such as long non-coding RNAs (lncRNA) and circular RNAs (circRNA), represent an unexplored resource for identifying new contributors to LUAD-BM [11].

Results from KEGG pathway analysis indicated that these genes were enriched in pathways involving cell adhesion molecules, chemokine signaling, cytokine–cytokine receptor interaction, and differentiation pathways of Th1, Th2, and Th17 cells (Appendix A). It is important to note that the most significantly altered immune-related pathway found was the cytokine–cytokine receptor interaction, which included 11/102 differentially expressed genes associated with BM from LUAD (*IL2RB*, *CCL5*, *CD27*, *CXCL13*, *CXCL9*, *CCL14*, *CXCL10*, *CCL18*, *CCL19*, *CCR7*, and *LTB*). Cytokine–cytokine receptor interactions can regulate immune responses by activating or inhibiting immune cells, including T cells, B cells, and natural killer (NK) cells [77]. These data suggest tissue specificity in the expression of some genes in BM and the regulation of pathways mainly related to the immune system. Similar findings were reported by Tsakonas et al., who identified a pattern of decreased gene expression in BM of NSCLC-related genes primarily involved in immune response, immune cell activation, and cytokine and chemokine receptors [78]. Previous studies have demonstrated that inflammatory chemokines and their receptors regulate tumor cell migration and participate in tumor growth, metastasis, angiogenesis, and invasion through the interaction between mesenchymal cells and neoplastic cells [79,80].

To explore the specific characteristics of LUAD-BM, we analyzed the 102 genes related to BM in the specific context of the disease using the DO database and their cancer-specific relationship using the Network of Cancer Genes database. Our results showed no significant associations with specific cancers. These data suggest that LUAD-BM are distinct entities compared to the primary tumor, as reported in previous studies [16,20,81].

Additionally, protein–protein interactions among the DEGs were predicted. The interaction with the highest combined score was between the CD3D and CD247 proteins (combined score = 0.999). Both proteins are part of the TCR-CD3 complex present on the surface of T lymphocytes, which plays an essential role in the adaptive immune response. When antigen-presenting cells (APCs) activate the T-cell receptor (TCR), signals mediated by the TCR are transmitted through the cell membrane by the CD3, CD3D, CD3E, CD3G, and CD3Z chains [82]. In addition to its signaling role in T-cell activation, CD3D plays an essential role in thymocyte differentiation by participating in the assembly and proper surface expression of the intracellular TCR-CD3 complex. In the absence of a functional TCR-CD3 complex, thymocytes are unable to differentiate properly. CD3D also interacts with CD4 and CD8 and thus serves to establish a functional link between the TCR and the CD4 and CD8 co-receptors, which is necessary for the activation and positive selection of CD4 or CD8 T cells [83]. The TCR-CD3 complex represents a promising avenue for immunotherapy in metastatic brain cancer. The potential benefits of TCR-CD3-based interventions include potentiation of antigen recognition, immune activation, and immunosuppression reversal (for example, by providing T-cell-activating stimuli). There are several preclinical and clinical uses of CD3 modulators that may benefit patients suffering from brain metastasis [84].

After constructing the PPI network, we built a co-expression network in which we identified the top 20 elements of the network, also known as ‘hub’ genes. Of the 20 ‘hub’ genes, the CD69 gene had the highest degree (score = 396.192). The protein encoded by this gene is involved in lymphocyte proliferation and functions as a signal transduction receptor in lymphocytes, NK cells, and platelets. It also regulates the differentiation of regulatory T cells (Tregs) as well as the secretion of IFN-γ, IL-17, and IL-22 [85]. These results support the hypothesis that the immune system plays a significant role in the development of LUAD-BM and suggests that targeting the immune system may be a promising approach for the treatment and management of LUAD-BM.

Previous studies have provided compelling evidence for the involvement of the immune system in the development of BM as reviewed by Leibold et al. [86]. It is well-established that the immune system plays a crucial role in regulating various stages of cancer progression, including the development and dissemination of metastatic tumors [87,88]. In the context of BM, immune cells and their interactions with cancer cells and the tumor microenvironment have been shown to have significant implications for progression [89]. Cytokines, chemokines, and growth factors play critical roles in the complex interplay between cancer cells and their surrounding environment during the development and progression of BM [90]. Recent studies have focused on identifying immunological characteristics specific to BM from NSCLC. Kudo et al. conducted a comparative immune gene profiling analysis and demonstrated elevated infiltration of M2 macrophages in BM compared to paired NSCLC samples [91]. Zhang et al. observed increased expression of CD163 M2 macrophages in the tumor brain microenvironment, which was correlated with a significant promotion of neo-angiogenesis [92]. Furthermore, Berghoff et al. found notable differences in the infiltration patterns of microglia and M2 macrophages between BM originating from NSCLC and melanoma [93]. Song et al. examined the expression of 770 genes related to the immune system across 28 different tissues, including primary tumors and BM of NSCLC. Utilizing the NanoString, Seattle, Washington, United States, nCounter PanCancer Immune Profiling Panel, they discovered that BM from EGFR-mutated adenocarcinoma exhibited increased activation of various immune-related pathways when compared to EGFR-wild-type adenocarcinoma. However, these same pathways were not observed in the primary tumors [94]. Additionally, the study discovered that the majority of immune cell subsets were reduced in BM in comparison to primary tumors. The reduction in immune cell subsets suggests the existence of possible immunosuppressive mechanisms within the environment of BM [94]. Recently, Najjary et al. used a combined approach based on NanoString’s nCounter, immunohistochemistry, and the GeoMx™ Digital Spatial Profiler (DSP) to demonstrate a more extensive infiltration of immune cells in BM from lung adenocarcinoma compared to BM from breast cancer [95]. Furthermore, the authors confirmed the higher protein expression of immune-related targets in BM-LUAD (CD14, CD163, GZMA, BCL-6, BAD, BCLXL, 4-1BB, VISTA, and IDO1). Interestingly, the gene *GZMA* was identified as a ‘hub’ gene in the present study.

GZMA, a member of the serine protease family, is primarily found in the cytolytic granules of cytotoxic T lymphocytes (CTLs) and natural killer (NK) cells. It plays a significant role in cell-mediated cytotoxicity, which is a crucial immune response against tumor cells [96,97,98]. Previous studies have highlighted the importance of GZMA as a key effector molecule in regulatory T-cell function within the context of cancer [99,100]. Moreover, GZMA has been identified as a vital factor in inhibiting tumor growth, promoting apoptosis, and stimulating antigen-specific cytotoxic CD8+ T-lymphocytes [101]. Notably, Zhou et al. demonstrated that GZMA, derived from cytotoxic lymphocytes, specifically activates the gasdermin-B protein, contributing to the elimination of target cells [102].

Recent investigations by Huo et al. have revealed lower *GZMA* expression in breast cancer tissue compared to normal tissue. Additionally, a correlation has been observed between *GZMA* and T-cell checkpoints, including PD-1, PD-L1, and CTLA-4, in breast cancer [103]. Furthermore, quantitative immunofluorescence analysis has demonstrated a positive association between *GZMA* expression and the presence of dendritic cells and CD8+ T cells infiltrating breast cancer tissue. These findings suggest a positive association between *GZMA* expression and enhanced infiltration of dendritic and CD8+ T cells in breast cancer. In our study, we observed a significant positive correlation (*p* < 0.05) between *GZMA* expression and the infiltration of CD8 T cells and dendritic cells. Additionally, we found a positive correlation (*p* < 0.05) between CD8 T-cell infiltration and monocytes, which represent another subset of myeloid cells. Interestingly, *GZMA* expression was found to be associated with immune stimulators such as CD48 and CD27 [103]. These genes were identified as downregulated hub genes in our study. Based on these observations, we hypothesize that the downregulation of *GZMA* in LUAD-BM may play a crucial role in modulating immune cell infiltration and contribute to the establishment of a suppressive immune microenvironment.

Furthermore, our study revealed that the proportions of resting memory CD4 T cells comprised the largest cellular fraction of the total immune cells, accounting for 17.4% of the total immune cells in BM and 15.24% in primary tumors. Compared to primary tumors, the proportions of resting dendritic cells and neutrophils showed statistical significance by the Wilcoxon-Mann-Whitney test (*p* < 0.05). Resting DCs were reduced in BM compared to primary tumors, while neutrophils showed an increased fraction. DCs are known for their essential role in activating the anti-tumor immune response through phagocytosis and the presentation of antigens from apoptotic tumor cells to CD4+ and CD8+ T cells. Normally, DCs are not found in the normal brain parenchyma but are present in vascular-rich compartments such as the choroid plexus and meninges [104]. In the context of pathological conditions such as cancer, DCs can migrate to the brain through afferent lymphatic vessels or endothelial venules [105].

Supporting our findings, Kim et al. also demonstrated the presence of DCs in LUAD-BM using scRNA-seq analysis [106]. Specifically, they identified CD163 + CD14+ DCs as the predominant subset in LUAD-BM [106]. Notably, CD163 + CD14+ DCs were found to be abundant in early- and advanced-stage lung cancer primary tissues but less abundant in metastatic lymph nodes and LUAD-BM [106]. DCs play a crucial role in the immune response by recognizing pathogens, coordinating both innate and adaptive immune responses, and secreting inflammatory mediators. DCs are unique in their ability to activate and direct naive T cells towards various effector cell types, such as Th1, Th2, Th17, and Tregs, depending on the specific cytokine and costimulatory signals they provide [107,108]. CD163 + CD14+ DC subset has been shown to possess a strong Th17 polarizing capacity, as evidenced by the pro-Th17 gene signature [109], which was consistent with our results (Figure 9E). Interestingly, our analysis revealed a lower fraction of DCs in BM compared to the primary tumor. This decrease in DC abundance within the BM microenvironment suggests the presence of an immunosuppressive environment that may have implications for the optimal presentation of tumor antigens in LUAD-BM. Furthermore, the absence of activated DCs in the BM microenvironment further supports the notion of an immunosuppressive setting. These findings highlight the possibility of sub-optimal tumor antigen presentation within LUAD-BM, potentially impairing the generation of effective anti-tumor immune responses. The immunosuppressive microenvironment observed in BM may contribute to the evasion of immune surveillance and facilitate tumor progression. However, further investigation is necessary to elucidate the underlying mechanisms responsible for the observed decrease in DCs abundance and the absence of activated DCs in the BM microenvironment.

Furthermore, our findings support the notion that polymorphonuclear myeloid-derived suppressor cells (PMN-MDSCs) may play a role in creating an immunosuppressive microenvironment. PMN-MDSCs have emerged as a distinct population of myeloid cells with immunosuppressive properties [110]. In the context of cancer, PMN-MDSCs have been implicated in the establishment of an immunosuppressive microenvironment that facilitates tumor growth and inhibits anti-tumor immune responses [110]. These cells have the ability to suppress the activity of various immune cells, including T cells, natural killer cells, and dendritic cells, thereby impairing the host’s capacity to mount an effective immune response against cancer cells. Additionally, PMN-MDSCs contribute to tumor progression by promoting angiogenesis, tissue remodeling, and metastasis [110]. In line with our observations, Sun et al. have also identified the presence of PMN-MDSCs in gliomas and lung cancer brain metastases [111]. Notably, the authors demonstrated a high expression of L-selectin in PMN-MDSCs, which has been reported to regulate human neutrophil transendothelial migration [112].

Previous studies have shed light on the immune cell composition within primary brain tumors, with macrophages being identified as the predominant immune cell type, often constituting up to 30% of the tumor mass [113,114,115]. Wang et al. specifically demonstrated an increase in type-2 (M2) polarized macrophages in mesenchymal gliomas [116]. Furthermore, Liang et al. showed that neutrophils contribute to glioblastoma progression by supporting the expansion of the glioma stem cell pool through a S100 protein-dependent mechanism [117]. S100 proteins have been associated with the dissemination of breast cancer and are upregulated in the premetastatic brain, promoting neutrophil recruitment and subsequent metastatic seeding [118,119]. Interestingly, our observations in LUAD-BM align with these findings, suggesting potential common immune microenvironment features between BM and primary brain tumors. As a result, there may be opportunities for immunotherapeutic strategies, such as targeting tumor-associated neutrophils (TANs), that could be applicable to both LUAD-BM and primary brain tumors. Moreover, significant progress in the development of efficient delivery methods for immunotherapy, including nanocell-based drug delivery systems and drug repurposing [120], reinforces the potential of utilizing immunotherapy in treating LUAD-BM as well as primary brain tumors.

Overall, our study provides valuable insights into the complex immune microenvironment of LUAD-BM. The identified genes and pathways offer potential targets for therapeutic interventions aimed at overcoming immunosuppression and improving patient outcomes. However, further investigations are needed to elucidate the underlying mechanisms responsible for the observed changes in immune cell subsets and the immunosuppressive microenvironment in BM. Additionally, exploring the role of PMN-MDSCs could provide further insights into their contribution to the immunosuppressive tumor microenvironment.

## 5. Conclusions

In conclusion, we identified 102 genes with altered expression levels related to LUAD-BM, with most showing decreased expression in BM. Pathway analysis revealed enrichment in genes involved in cell adhesion molecules, chemokine signaling, cytokine–cytokine receptor interaction, and differentiation pathways of Th1, Th2, and Th17 cells. Our analysis also identified key ‘hub’ genes, including *CD69* and *GZMA*, which are involved in lymphocyte proliferation, immune cell activation, and cytokine regulation. We found that the downregulation of *GZMA* in LUAD-BM may contribute to the establishment of a suppressive immune microenvironment. Furthermore, we observed alterations in immune cell populations in the brain metastatic microenvironment, including a decrease in dendritic cells and an increase in neutrophils, indicating the presence of an immunosuppressive environment within LUAD-BM.

Our findings highlight the importance of the immune system in the development and progression of LUAD-BM. Targeting the immune system may hold promise as a therapeutic approach for the treatment and management of BM in patients with adenocarcinoma. Further investigation is warranted to elucidate the underlying mechanisms driving the immune system dysregulation in LUAD-BM and to explore potential immunotherapeutic strategies to improve patient outcomes.

## Figures and Tables

**Figure 1 cancers-15-04526-f001:**
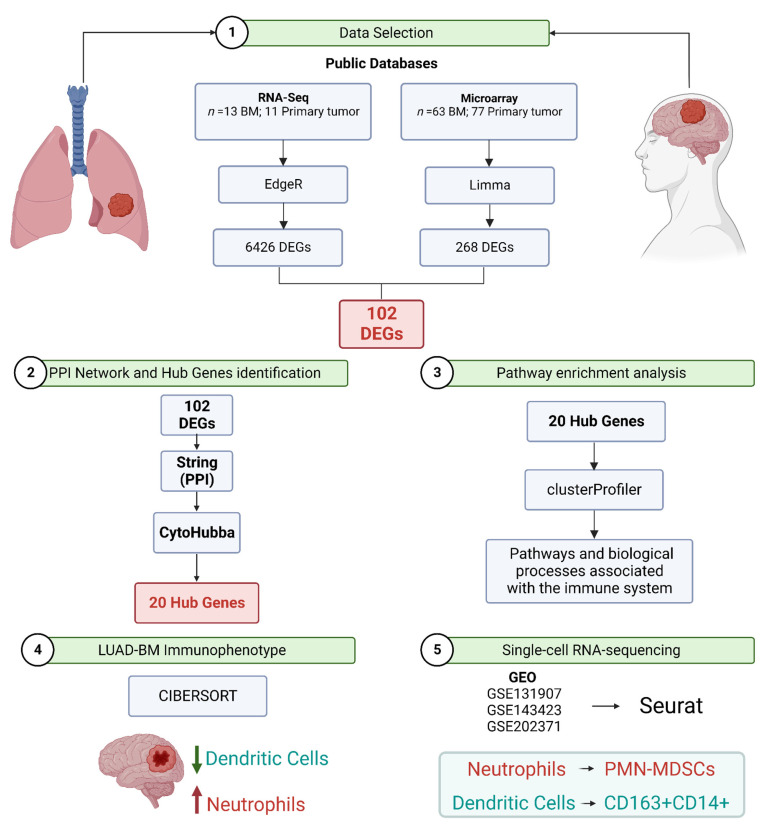
The experimental design and the significant findings of this study. BM: Brain metastasis; DEGs: Differentially expressed genes; PMN-MDSCs: Polymorphonuclear myeloid-derived suppressor cells; PPI: Protein–protein interaction.

**Figure 2 cancers-15-04526-f002:**
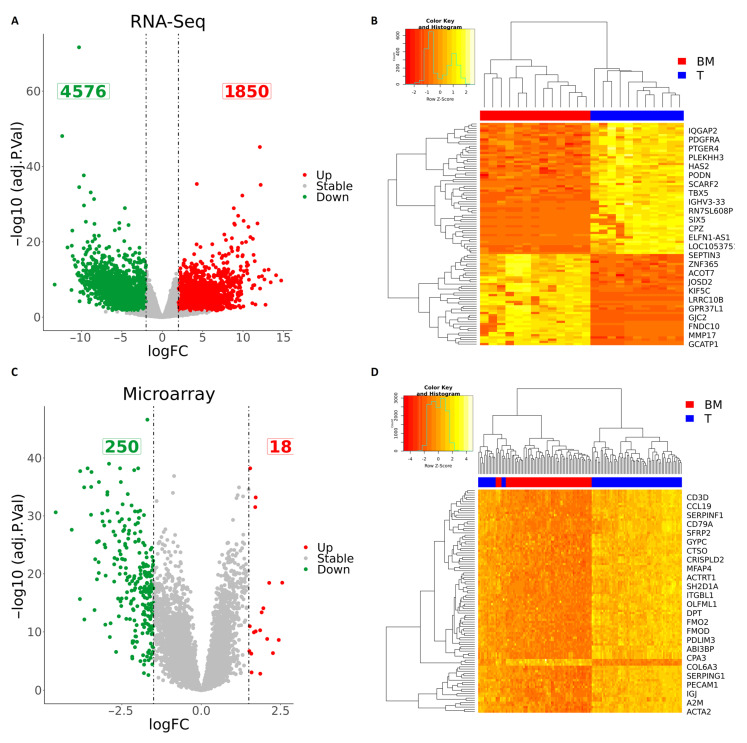
(**A**) Volcano plot indicating differentially expressed genes (DEGs) in brain metastases (BM) compared to the primary tumor (T) in RNA-seq data (red and green colors indicate |logFC| > 2 and adj. *p* < 0.05; other genes are colored gray. (**B**) Hierarchical cluster showing the expression profile of the top 100 DEGs in the RNA-seq data. (**C**) Volcano plot indicating differentially expressed genes (DEGs) in brain metastases (BM) in comparison to the primary tumor (T) in the microarray data (red and green colors indicate |logFC| > 1.5 and adj. *p* < 0.05; other genes are colored gray. (**D**) Hierarchical clustering showing the expression profile of the top 100 DEGs on the microarray data.

**Figure 3 cancers-15-04526-f003:**
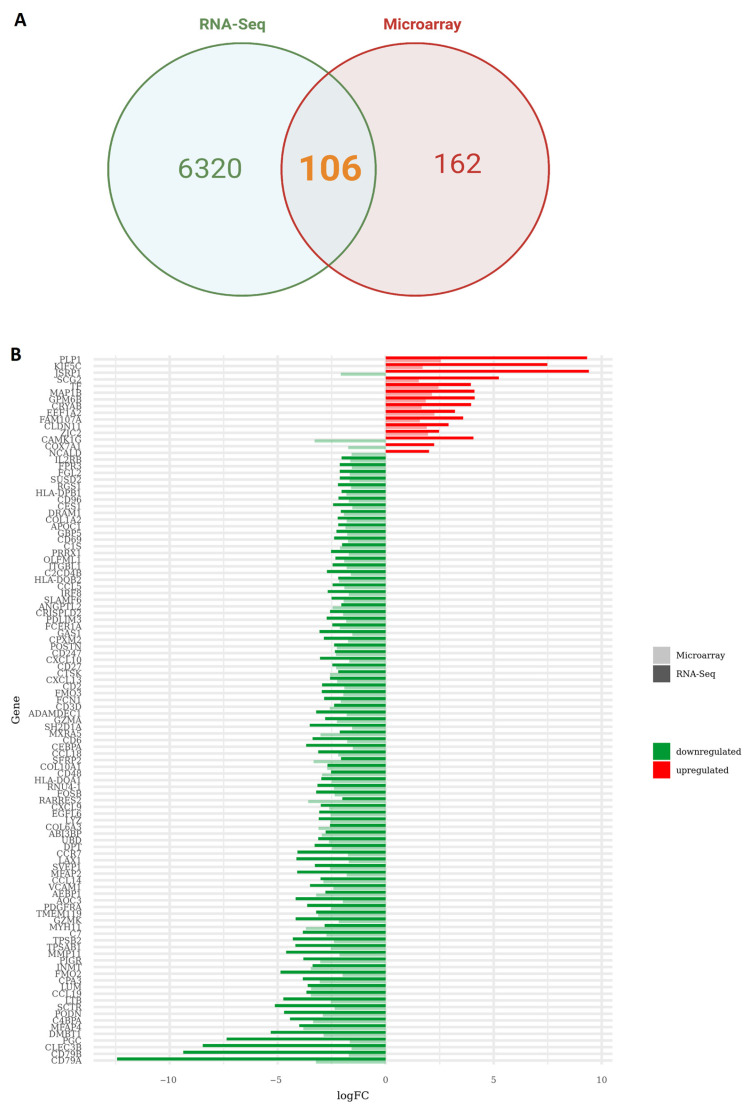
(**A**) Venn diagram showing the overlap between differentially expressed genes (DEGs) between RNA-seq and microarray data. (**B**) Bar graph indicating logFC values and expression direction of the 106 DEGs superimposed across sequencing technologies.

**Figure 4 cancers-15-04526-f004:**
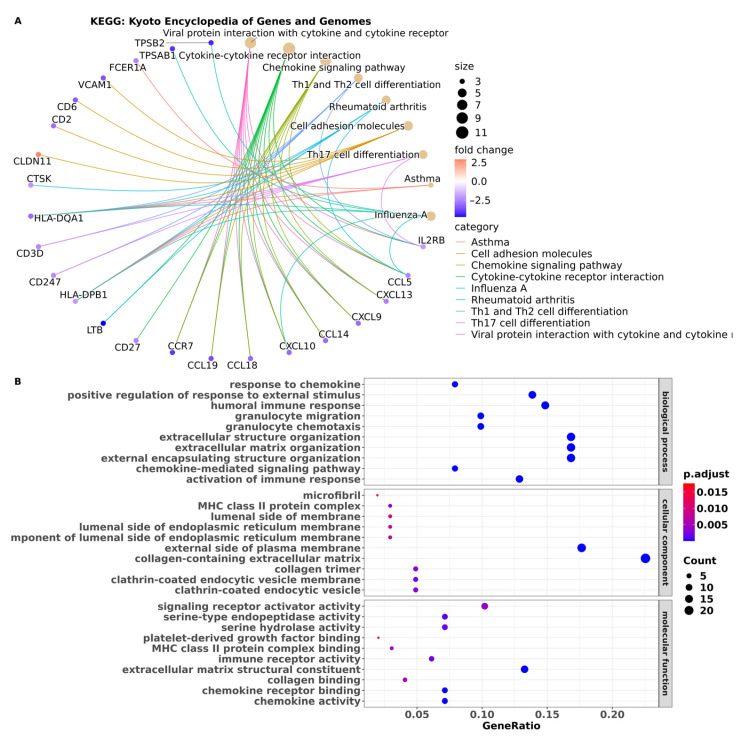
Functional annotation and enrichment analysis. (**A**) Enrichment analysis interaction network from the Kyoto Encyclopedia of Genes and Genomes (KEGG). The node size represents the number of genes according to each KEGG category, and the color of the nodes represents the logFC value per gene within each enriched KEGG category, as shown by the legend. Borders highlight interactions between the KEGG category and the genes that enrich it. (**B**) Enrichment dot plot of the term Genetic Ontology. The graph presents the top 10 enriched ontologies for each of the instance terms (biological process, molecular function, and cellular component) with adj. *p*-value < 0.05. The *X*-axis presents the number of genes that enrich the ontology term, and the point size is proportional to this number.

**Figure 5 cancers-15-04526-f005:**
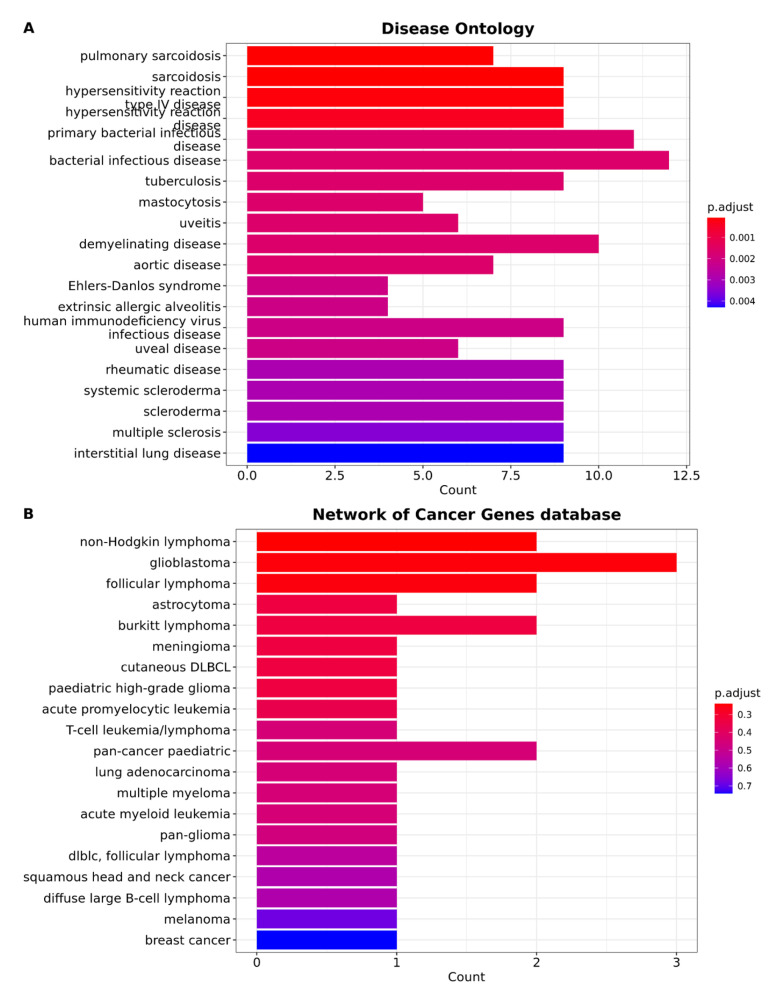
Gene set enrichment analysis. (**A**) Bar plot of Disease Ontology enrichment analysis. The plot was created using clusterProfiler and Disease Ontology annotations from the DO database. The *x*-axis represents the number of genes enriching the ontology term, and the color of the bars represents the adjusted *p*-value. (**B**) Bar plot of enrichment analysis based on the Network of Cancer Genes database. The *x*-axis represents the number of genes enriching the ontology term, and the color of the bars represents the adjusted *p*-value.

**Figure 6 cancers-15-04526-f006:**
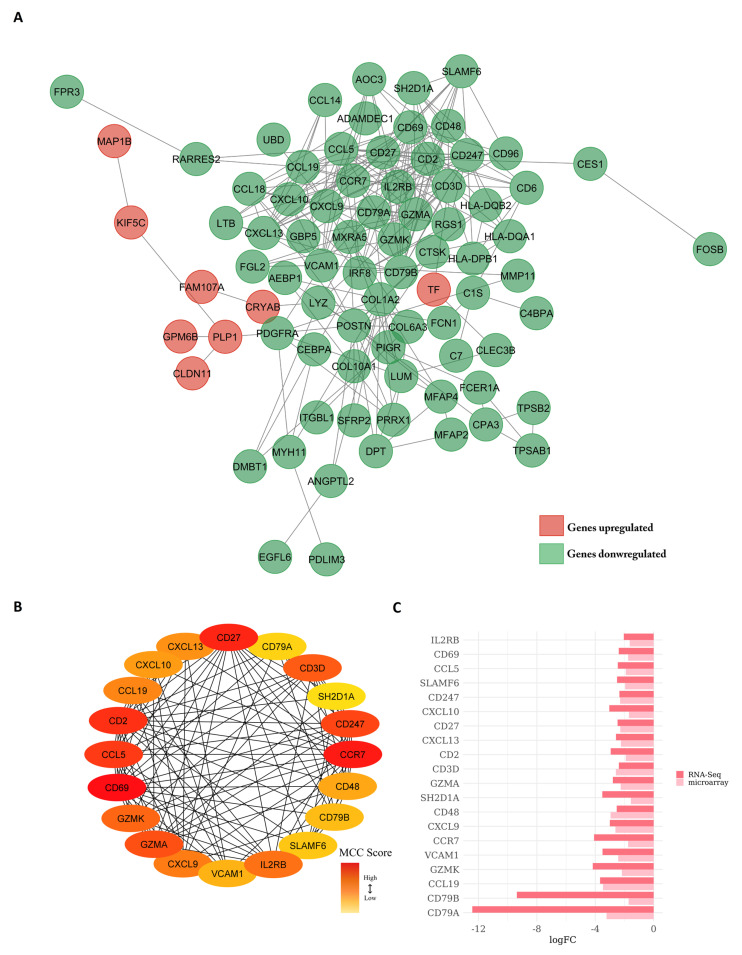
(**A**) Protein–protein interaction network (PPI) visualized with Cytoscape. The nodes represent the proteins. Borders highlight interactions between proteins. Upregulated genes are marked in red and downregulated genes are marked in green. (**B**) ‘Hub’ genes. The color red to yellow represents the degree of connectivity from top to bottom. (**C**) Bar chart indicating logFC values and expression direction of the 20 ‘hub’ genes.

**Figure 7 cancers-15-04526-f007:**
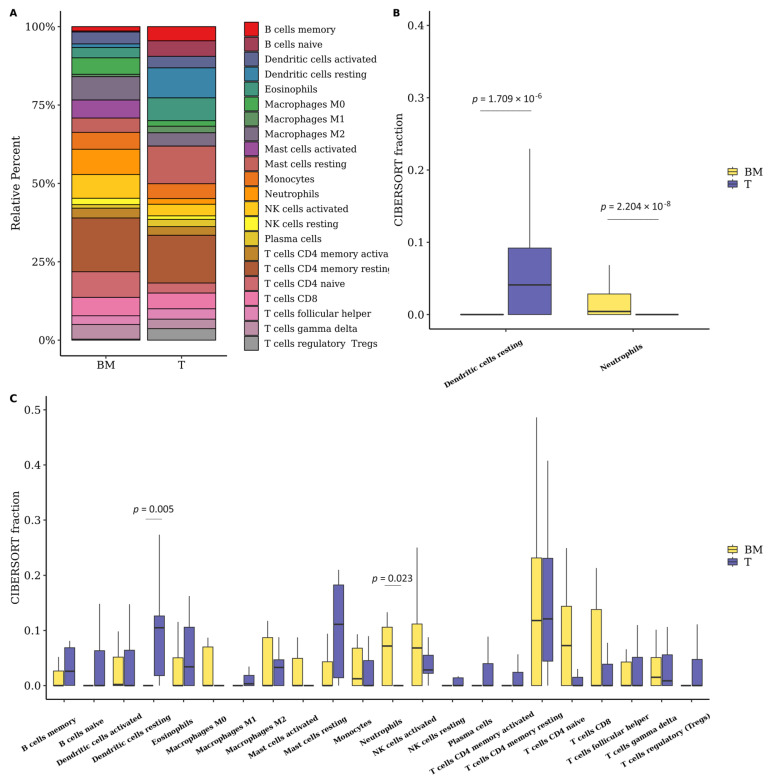
(**A**) Infiltrating immune cell composition in brain metastasis (BM) and primary tumor (T) is summarized from mean values calculated for each group. The bar graph shows the difference between CIBERSORTx immune cell fractions between brain metastases (BM) and primary tumors (T). (**B**) Results were generated using microarray data. (**C**) Results were generated using RNA-seq data.

**Figure 8 cancers-15-04526-f008:**
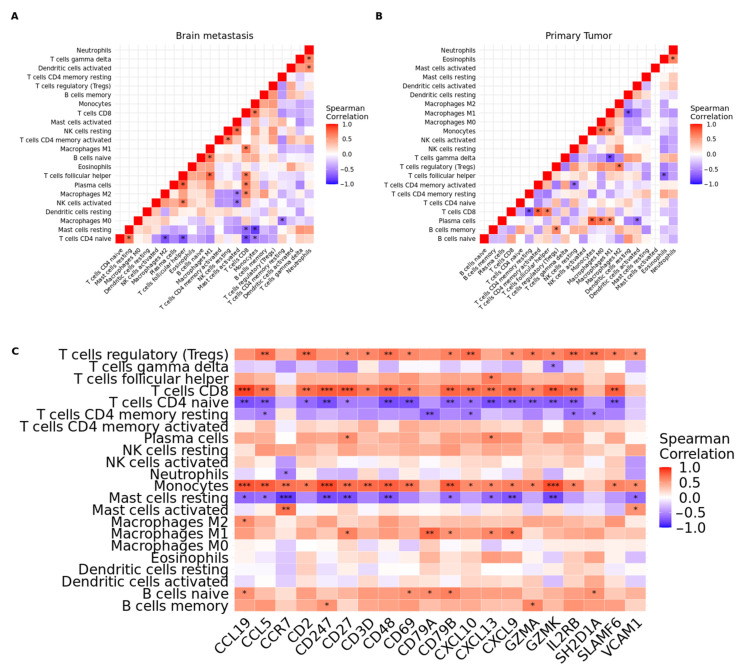
Correlation matrix of all 22 immunological proportions. (**A**) Brain metastasis. (**B**) Primary tumor. * represents significant correlations (*p* < 0.05). (**C**) Correlation plot (Spearman correlation coefficients) of ‘hub’ gene expression and proportion of infiltrating immune cells in brain metastasis. Colors in the heatmap indicate the strength of the correlation. Asterisks indicate the level of significance (* *p* < 0.05, ** *p* < 0.01, *** *p* < 0.001).

**Figure 9 cancers-15-04526-f009:**
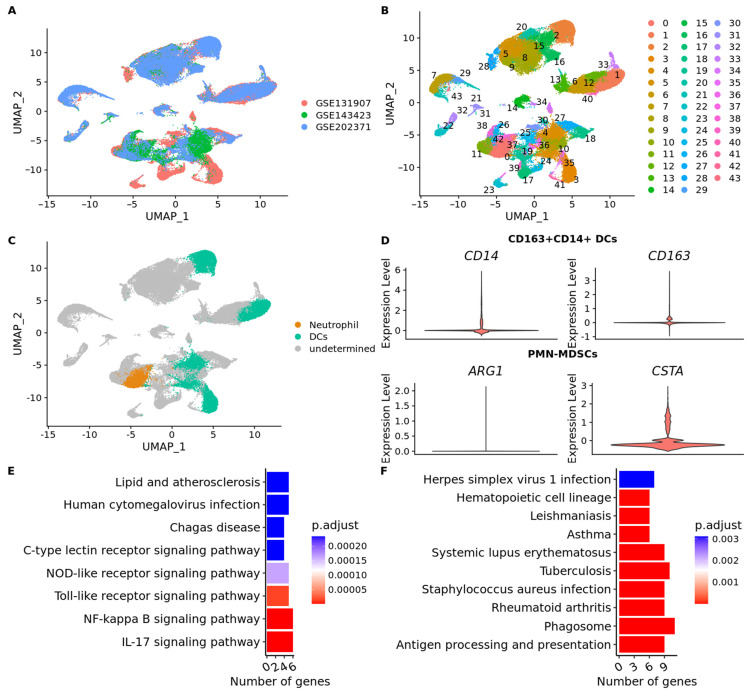
(**A**) The distribution of cells in each cluster according to the dataset. (**B**) 57,774 cells were divided into 43 separate clusters. (**C**) After annotation of cell type dendritic cells and neutrophils were found. (**D**) Violin plots of the expression of marker genes of neutrophil and dendritic cell subtypes. The violin plots to all the markers are shown in Appendix A. (**E**) Bar chart of enrichment analysis based on KEGG database for PMN-MDSCs. The *X*-axis shows the number of genes that enrich the pathway and the color of the bars represents the adjusted *p* value. (**F**) Bar chart of enrichment analysis based on KEGG database for CD163 + CD14+ dendritic cells. The *X*-axis shows the number of genes that enrich the pathway, and the color of the bars represents the adjusted *p* value.

**Table 1 cancers-15-04526-t001:** Description of publicly available transcriptomic data used in this meta-analysis.

Study	Database	Access	Technology	Platform	Tissue	Number of Samples	Reference
1	EGA	EGAS00001004078	RNA-seq	Illumina HiSeq 2000	BM/T	5 BM 4 T	[68]
2	EGA	EGAS00001004006	RNA-seq	Illumina HiSeq X Ten	T	7	[69,70]
3	SRA	PRJNA510710	RNA-seq	Illumina HiSeq 2500	BM	2	[71]
4	GEO	GSE141685	RNA-seq	Illumina HiSeq X Ten	BM	6	NA
5	ArrayExpress	E-MTAB-8659	microarray	Illumina HumanHT-12 V4.0 expression beadchip	BM	63	NA
6	GEO	GSE60645	microarray	Illumina HumanHT-12 V4.0 expression beadchip	T	77	[72]

EGA: European Genome-phenome Archive; GEO: Gene Expression Omnibus; BM: Brain metastasis; T: Primary tumor. NA: Not available.

## Data Availability

The datasets that support the findings of this study are available in the European Genome-phenome Archive (EGA) under accession numbers EGAS00001004078 and EGAS00001004006; Sequence Read Archive (SRA) data under accession number PRJNA510710; Gene Expression Omnibus (GEO) under accession numbers GSE141685, GSE60645, GSE131907, GSE143423, and GSE202371; and the functional genomics data collection (ArrayExpress) under accession number E-MTAB-8659.

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
