# Peer review of "Identifying New Contributors to Brain Metastasis in Lung Adenocarcinoma: A Transcriptomic Meta-Analysis"

_cancers, 2023, doi:10.3390/cancers15184526_

Round 1
Reviewer 1 Report
1. Author Mentioned identified “102 genes with significantly deregulated expression levels in BM tissues” – are there any more identified genes to date?
2. "18/20 ‘hub’ genes (CD69, CCR7, CD27, CD2, CCL5, CD247, GZMA, CD3D, GZMK, IL2RB, CXCL9, CCL19, CXCL13, CXCL10, CD48, VCAM1, CD79B and SLAMF6), followed by CD8 T cell infiltration correlated with expression of 16/20 ‘hub’ genes (CD69, CD27, CD2, CCL5, CD247, GZMA, CD3D, GZMK, IL2RB, CXCL9, CCL19, CXCL13, CXCL10, CD48, CD79B and SLAMF6)." - It would be good to present genes/hub in a table
3. In the discussion section the authors mention immune response, but so far the research is still going on to find the best delivery method for oncogenic immunosuppression biomarkers for therapeutic targets of BM and LUAD management.
4. Authors mentions "The interaction with the highest combined score was between the CD3D-CD247 proteins (combined score = 0.999). Both proteins are part of the TCR-CD3 complex present on the surface of T lymphocytes, which plays an essential role in the adaptive immune response." - So, how this kind of study will be beneficial for patients suffering from Metastatic brain cancer? Moreover, how to deal with off target effects of miRNA?
5. Did the authors find any lncRNA or circRNA associated with BM-LUAD in their analysis? It is not clear.
6. Authors have indicated that these pathways in BM are associated with the immune system, but again these pathways can also be important for normal cells. Which pathway was most significantly altered, in response to the alteration in the genes and can be associated with the immune system in BM?
The article is clear but the grammar needs improvement. All the data is shared, and overall the article structure is good.
Reviewer 2 Report
BM’s remain a main challenge in lung cancer. Using different sequencing technologies, the authors aim at identifying transcriptomic changes that were specific to metastatic tumor cells within the brain. Data comes from different public repositories and databases.
This is a good approach and even though these repositories are growing in the end a limited number of BM/non-BM sets were identified.
The manuscript is extremely long (29p) but it is also well written and contains a wealth of information. I have tried to see if it could be shortened or divided in 2 papers, but it actually makes sense to have as 1 paper and although lots of detail about the analysis may tire the average reader it gives valuable information about the analysis.
Figures are illustrative and helps with understanding although some of the volcano plots could probably be moved to an appendix.
The discussion is also very long but addresses the analysis performed so also here difficult to shorten the text.
My only suggestion is to highlight that the samples analyzed is pairs of BM and non-BM samples from same patient as this is a major strength of the analysis.
Reviewer 3 Report
The manuscript presents a very comprehensive metaanalysis of available transcriptomic datasets in order to identify new contributors to BM in LUAD. The overview is presented in a very clear and indicates interesting results that may shade the way for future clinical and experimental studies.
